# One-Sided Unsupervised Domain Mapping

Sagie Benaim[1] and Lior Wolf[1,2]

[1]The Blavatnik School of Computer Science , Tel Aviv University, Israel
[2]Facebook AI Research

## Abstract

In unsupervised domain mapping, the learner is given two unmatched datasets $A$ and $B$. The goal is to learn a mapping $G_{AB}$ that translates a sample in $A$ to the analog sample in $B$. Recent approaches have shown that when learning simultaneously both $G_{AB}$ and the inverse mapping $G_{BA}$, convincing mappings are obtained. In this work, we present a method of learning $G_{AB}$ without learning $G_{BA}$. This is done by learning a mapping that maintains the distance between a pair of samples. Moreover, good mappings are obtained, even by maintaining the distance between different parts of the same sample before and after mapping. We present experimental results that the new method not only allows for one sided mapping learning, but also leads to preferable numerical results over the existing circularity-based constraint. Our entire code is made publicly available at `https://github.com/sagiebenaim/DistanceGAN`.

## 1 Introduction

The advent of the Generative Adversarial Network (GAN) [6] technology has allowed for the generation of realistic images that mimic a given training set by accurately capturing what is inside the given class and what is "fake". Out of the many tasks made possible by GANs, the task of mapping an image in a source domain to the analog image in a target domain is of a particular interest.

The solutions proposed for this problem can be generally separated by the amount of required supervision. On the one extreme, fully supervised methods employ pairs of matched samples, one in each domain, in order to learn the mapping [9]. Less direct supervision was demonstrated by employing a mapping into a semantic space and requiring that the original sample and the analog sample in the target domain share the same semantic representation [22].

If the two domains are highly related, it was demonstrated that just by sharing weights between the networks working on the two domains, and without any further supervision, one can map samples between the two domains [21, 13]. For more distant domains, it was demonstrated recently that by symmetrically leaning mappings in both directions, meaningful analogs are obtained [28, 11, 27]. This is done by requiring circularity, i.e., that mapping a sample from one domain to the other and then back, produces the original sample.

In this work, we go a step further and show that it is possible to learn the mapping between the source domain and the target domain in a one-sided unsupervised way, by enforcing high cross-domain correlation between the matching pairwise distances computed in each domain. The new constraint allows one-sided mapping and also provides, in our experiments, better numerical results than circularity. Combining both of these constraints together often leads to further improvements.

Learning the new constraint requires comparing pairs of samples. While there is no real practical reason not to do so, since training batches contain multiple samples, we demonstrate that similar constraints can even be applied per image by computing the distance between, e.g., the top part of the image and the bottom part.

## 1.1 Related work

**Style transfer** These methods [5, 23, 10] typically receive as input a style image and a content image and create a new image that has the style of the first and the content of the second. The problem of image translation between domains differs since when mapping between domains, part of the content is replaced with new content that matches the target domain and not just the style. However, the distinction is not sharp, and many of the cross-domain mapping examples in the literature can almost be viewed as style transfers. For example, while a zebra is not a horse in another style, the horse to zebra mapping, performed in [28] seems to change horse skin to zebra skin. This is evident from the stripped Putin example obtained when mapping the image of shirtless Putin riding a horse.

**Generative Adversarial Networks** GAN [6] methods train a generator network $G$ that synthesizes samples from a target distribution, given noise vectors, by jointly training a second network $D$. The specific generative architecture we and others employ is based on the architecture of [18]. In image mapping, the created image is based on an input image and not on random noise [11, 28, 27, 13, 22, 9].

**Unsupervised Mapping** The work that is most related to ours, employs no supervision except for sample images from the two domains. This was done very recently [11, 28, 27] in image to image translation and slightly earlier for translating between natural languages [24]. Note that [11] proposes the "GAN with reconstruction loss" method, which applies the cycle constraint in one side and trains only one GAN. However, unlike our method, this method requires the recovery of both mappings and is outperformed by the full two-way method.

The CoGAN method [13], learns a mapping from a random input vector to matching samples from the two domains. It was shown in [13, 28] that the method can be modified in order to perform domain translation. In CoGAN, the two domains are assumed to be similar and their generators (and GAN discriminators) share many of the layers weights, similar to [21]. As was demonstrated in [28], the method is not competitive in the field of image to image translation.

**Weakly Supervised Mapping** In [22], the matching between the source domain and the target domain is performed by incorporating a fixed pre-trained feature map $f$ and requiring $f$-constancy, i.e, that the activations of $f$ are the same for the input samples and for mapped samples.

**Supervised Mapping** When provided with matching pairs of (input image, output image) the supervision can be performed directly. An example of such method that also uses GANs is [9], where the discriminator $D$ receives a pair of images where one image is the source image and the other is either the matching target image ("real" pair) or a generated image ("fake" pair); The linking between the source and the target image is further strengthened by employing the U-net architecture [19].

**Domain Adaptation** In this setting, we typically are given two domains, one having supervision in the form of matching labels, while the second has little or no supervision. The goal is to learn to label samples from the second domain. In [3], what is common to both domains and what is distinct is separated thus improving on existing models. In [2], a transformation is learned, on the pixel level, from one domain to another, using GANs. In [7], an unsupervised adversarial approach to semantic segmentation, which uses both global and category specific domain adaptation techniques, is proposed.

## 2 Preliminaries

In the problem of unsupervised mapping, the learning algorithm is provided with unlabeled datasets from two domains, $A$ and $B$. The first dataset includes i.i.d samples from the distribution $p_A$ and the second dataset includes i.i.d samples from the distribution $p_B$. Formally, given

$$\{x_i\}_{i=1}^m \text{ such that } x_i \overset{\text{i.i.d}}{\sim} p_A \quad \text{and} \quad \{x_j\}_{j=1}^n \text{ such that } x_j \overset{\text{i.i.d}}{\sim} p_B,$$

our goal is to learn a function $G_{AB}$, which maps samples in domain $A$ to analog samples in domain $B$, see examples below. In previous work [11, 28, 27], it is necessary to simultaneously recover a second function $G_{BA}$, which similarly maps samples in domain $B$ to analog samples in domain $A$.

**Justification** In order to allow unsupervised learning of one directional mapping, we introduce the constraint that pairs of inputs $x, x'$, which are at a certain distance from each other, are mapped to pairs of outputs $G_{AB}(x), G_{AB}(x')$ with a similar distance, i.e., that the distances $\|x - x'\|$ and

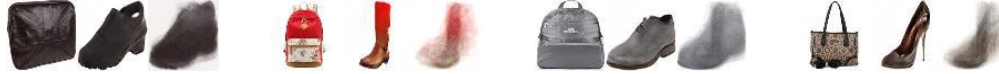

Figure 1: Each triplet shows the source handbag image, the target shoe as produced by Cycle-GAN's [28] mapper $G_{AB}$ and the results of approximating $G_{AB}$ by a fixed nonnegative linear transformation $T$, which obtains each output pixel as a linear combination of input pixels. The linear transformation captures the essence of $G_{AB}$ showing that much of the mapping is achieved by a fixed spatial transformation.

$\|G_{AB}(x) - G_{AB}(x')\|$ are highly correlated. As we show below, it is reasonable to assume that this constraint approximately holds in many of the scenarios demonstrated by previous work on domain translation. Although approximate, it is sufficient, since as was shown in [21], mapping between domains requires only little supervision on top of requiring that the output distribution of the mapper matches that of the target distribution.

Consider, for example, the case of mapping shoes to edges, as presented in Fig. 4. In this case, the edge points are simply a subset of the image coordinates, selected by local image criterion. If image $x$ is visually similar to image $x'$, it is likely that their edge maps are similar. In fact, this similarity underlies the usage of gradient information in the classical computer vision literature. Therefore, while the distances are expected to differ in the two domains, one can expect a high correlation.

Next, consider the case of handbag to shoe mapping (Fig. 4). Analogs tend to have the same distribution of image colors in different image formations. Assuming that the spatial pixel locations of handbags follow a tight distribution (i.e., the set of handbag images share the same shapes) and the same holds for shoes, then there exists a set of canonical displacement fields that transform a handbag to a shoe. If there was one displacement, which would happen to be a fixed permutation of pixel locations, distances would be preserved. In practice, the image transformations are more complex.

To study whether the image displacement model is a valid approximation, we learned a nonnegative linear transformation $T \in \mathbb{R}_+^{64^2 \times 64^2}$ that maps, one channel at a time, handbag images of size $64 \times 64 \times 3$ to the output shoe images of the same size given by the CycleGAN method. $T$'s columns can be interpreted as weights that determine the spread of mass in the output image for each pixel location in the input image. It was estimated by minimizing the squared error of mapping every channel (R, G, or B) of a handbag image to the same channel in the matching shoe. Optimization was done by gradient descent with a projection to the space of nonnegative matrices, i.e., zeroing the negative elements of $T$ at each iteration.

Sample mappings by the matrix $T$ are shown in Fig. 1. As can be seen, the nonnegative linear transformation approximates CycleGAN's multilayer CNN $G_{AB}$ to some degree. Examining the elements of $T$, they share some properties with permutations: the mean sum of the rows is 1.06 (SD 0.08) and 99.5% of the elements are below 0.01.

In the case of adding glasses or changing gender or hair color (Fig 3), a relatively minor image modification, which does not significantly change the majority of the image information, suffices in order to create the desired visual effect. Such a change is likely to largely maintain the pairwise image distance before and after the transformation.

In the case of computer generated heads at different angles vs. rotated cars, presented in [11], distances are highly correlated partly because the area that is captured by the foreground object is a good indicator of the object's yaw. When mapping between horses to zebras [28], the texture of a horse's skin is transformed to that of the zebra. In this case, most of the image information is untouched and the part that is changed is modified by a uniform texture, again approximately maintaining pairwise distances. In Fig 2(a), we compare the $L1$ distance in RGB space of pairs of horse images to the distance of the samples after mapping by the CycleGAN Network [28] is performed, using the public implementation. It is evident that the cross-domain correlation between pairwise distances is high. We also looked at Cityscapes image and ground truth label pairs in Fig 2(c), and found that there is high correlation between the distances. This is the also the case in many other literature-based mappings between datasets we have tested and ground truth pairs.

While there is little downside to working with pairs of training images in comparison to working with single images, in order to further study the amount of information needed for successful alignment, we also consider distances between the two halves of the same image. We compare the $L_1$ distance

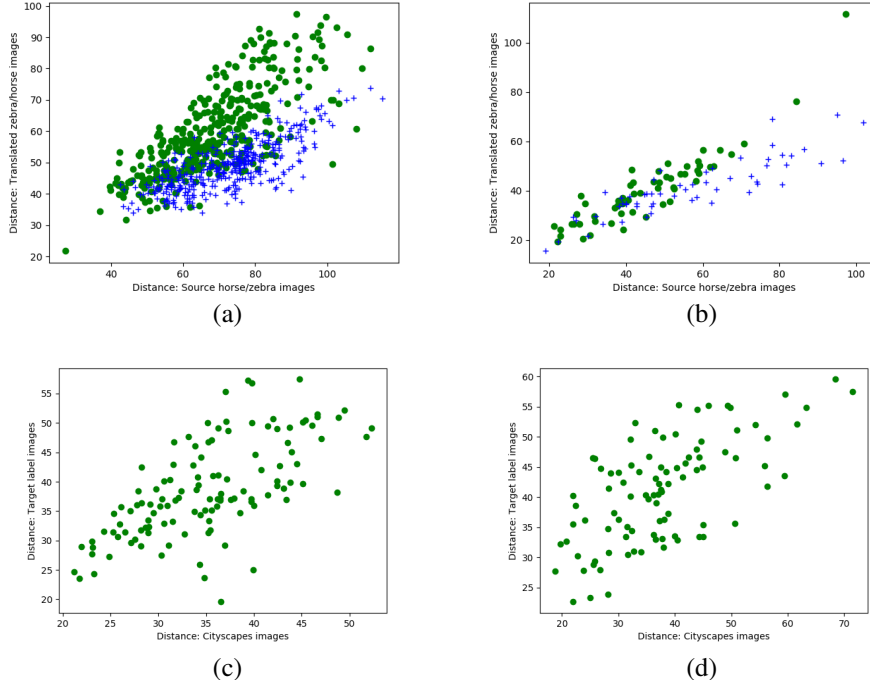

Figure 2: Justifying the high correlation between distances in different domains. (a) Using the CycleGAN model [28], we map horses to zebras and vice versa. Green circles are used for the distance between two random horse images and the two corresponding translated zebra images. Blue crosses are for the reverse direction translating zebra to horse images. The Pearson correlation for horse to zebra translation is $0.77$ (p-value $1.7e-113$) and for zebra to horse it is $0.73$ (p-value $8.0e-96$). (b) As in (a) but using the distance between two halves of the same image that is either a horse image translated to a zebra or vice-versa. The Pearson correlation for horse to zebra translation is $0.91$ (p-value $9.5e-23$) and for zebra to horse it is $0.87$ (p-value $9.7e-19$). (c) Cityscapes images and associated labels. Green circles are used for distance between two cityscapes images and the two corresponding ground truth images The Pearson correlation is $0.65$ (p-value $6.0e-16$). (d) As in (c) but using the distance between two halves of the same image. The Pearson correlation is $0.65$ (p-value $1.4e-12$).

between the left and right halves as computed on the input image to that which is obtained on the generated image or the corresponding ground truth image. Fig. 2(b) and Fig. 2(d) presents the results for horses to zebras translation and for Cityscapes image and label pairs, respectively. As can be seen, the correlation is also very significant in this case.

**From Correlations to Sum of Absolute Differences** We have provided justification and empirical evidence that for many semantic mappings, there is a high degree of correlations between the pairwise distances in the two domains. In other words, let $d_k$ be a vector of centered and unit-variance normalized pairwise distances in one domain and let $d'_k$ be the vector of normalized distances obtained in the other domain by translating each image out of each pair between the domains, then $\sum d_k d'_k$ should be high. When training the mapper $G_{AB}$, the mean and variance used for normalization in each domain are precomputed based on the training samples in each domain, which assumes that the post mapping distribution of samples is similar to the training distribution.

The pairwise distances in the source domain $d_k$ are fixed and maximizing $\sum d_k d'_k$ causes pairwise distances $d_k$ with large absolute value to dominate the optimization. Instead, we propose to minimize the sum of absolute differences $\sum_k |d_k - d'_k|$, which spreads the error in distances uniformly. The two losses $-\sum d_k d'_k$ and $\sum_k |d_k - d'_k|$ are highly related and the negative correlation between them was explicitly computed for simple distributions and shown to be very strong [1].

# 3   Unsupervised Constraints on the Learned Mapping

There are a few types of constraints suggested in the literature, which do not require paired samples. First, one can enforce the distribution of $G_{AB}(x) : x \sim p_A$, which we denote as $G_{AB}(p_A)$, to be indistinguishable from that of $p_B$. In addition, one can require that mapping from A to B and back would lead to an identity mapping. Another constraint suggested, is that for every $x \in B$ $G_{AB}(x) = x$. We review these constraints and then present the new constraints we propose.

**Adversarial constraints**   Our training sets are viewed as two discrete distributions $\hat{p}_A$ and $\hat{p}_B$ that are sampled from the source and target domain distributions $p_A$ and $p_B$, respectively. For the learned network $G_{AB}$, the similarity between the distributions $G_{AB}(p_A)$ and $p_B$ is modeled by a GAN. This involves the training of a discriminator network $D_B : B \to \{0, 1\}$. The loss is given by:

$$\mathcal{L}_{\text{GAN}}(G_{AB}, D_B, \hat{p}_A, \hat{p}_B) = \mathbb{E}_{x_B \sim \hat{p}_B}[\log D_B(x_B)] + \mathbb{E}_{x_A \sim \hat{p}_A}[\log(1 - D_B(G_{AB}(x_A)))]$$

This loss is minimized over $G_{AB}$ and maximized over $D_B$. When both $G_{AB}$ and $G_{BA}$ are learned simultaneously, there is an analog expression $\mathcal{L}_{\text{GAN}}(G_{BA}, D_A, \hat{p}_B, \hat{p}_A)$, in which the domains $A$ and $B$ switch roles and the two losses (and four networks) are optimized jointly.

**Circularity constraints**   In three recent reports [11, 28, 27], circularity loss was introduced for image translation. The rationale is that given a sample from domain $A$, translating it to domain $B$ and then back to domain $A$ should result in the identical sample. Formally, the following loss is added:

$$\mathcal{L}_{\text{cycle}}(G_{AB}, G_{BA}, \hat{p}_A) = \mathbb{E}_{x \sim \hat{p}_A} \|G_{BA}(G_{AB}(x)) - x\|_1$$

The $L1$ norm employed above was found to be mostly preferable, although $L2$ gives similar results. Since the circularity loss requires the recovery of the mappings in both directions, it is usually employed symmetrically, by considering $\mathcal{L}_{\text{cycle}}(G_{AB}, G_{BA}, \hat{p}_A) + \mathcal{L}_{\text{cycle}}(G_{BA}, G_{AB}, \hat{p}_B)$.

The circularity constraint is often viewed as a definite requirement for admissible functions $G_{AB}$ and $G_{BA}$. However, just like distance-based constraints, it is an approximate one. To see this, consider the zebra to horse mapping example. Mapping a zebra to a horse means losing the stripes. The inverse mapping, therefore, cannot be expected to recover the exact input stripes.

**Target Domain Identity**   A constraint that has been used in [22] and in some of the experiments in [28] states that $G_{AB}$ applied to samples from the domain $B$ performs the identity mapping. We did not experiment with this constraint and it is given here for completeness:

$$\mathcal{L}_{\text{T-ID}}(G_{AB}, \hat{p}_B) = \mathbb{E}_{x \sim \hat{p}_B} \|x - G_{AB}(x)\|_2$$

**Distance Constraints**   The adversarial loss ensures that samples from the distribution of A are translated to samples in the distribution of B. However, there are many such possible mappings. Given a mapping for $n$ samples of A to $n$ samples of B, one can consider any permutation of the samples in B as a valid mapping and, therefore, the space of functions mapping from A to B is very large. Adding the circularity constraint, enforces the mapping from B to A to be the inverse of the permutation that occurs from A to B, which reduces the amount of admissible permutations.

To further reduce this space, we propose a distance preserving map, that is, the distance between two samples in A should be preserved in the mapping to B. We therefore consider the following loss, which is the expectation of the absolute differences between the distances in each domain up to scale:

$$\mathcal{L}_{\text{distance}}(G_{AB}, \hat{p}_A) = \mathbb{E}_{x_i, x_j \sim \hat{p}_A} |\frac{1}{\sigma_A}(\|x_i - x_j\|_1 - \mu_A) - \frac{1}{\sigma_B}(\|G_{AB}(x_i) - G_{AB}(x_j)\|_1 - \mu_B)|$$

where $\mu_A, \mu_B$ ($\sigma_A, \sigma_B$) are the means (standard deviations) of pairwise distances in the training sets from $A$ and $B$, respectively, and are precomputed.

In practice, we compute the loss over pairs of samples that belong to the same minibatch during training. Even for minibatches with 64 samples, as in DiscoGAN [11], considering all pairs is feasible. If needed, for even larger mini-batches, one can subsample the pairs.

When the two mappings are simultaneously learned, $\mathcal{L}_{\text{distance}}(G_{BA}, \hat{p}_B)$ is similarly defined. In both cases, the absolute difference of the $L1$ distances between the pairs in the two domains is considered.

In comparison to circularity, the distance-based constraint does not suffer from the model collapse problem that is described in [11]. In this phenomenon, two different samples from domain A are mapped to the same sample in domain B. The mapping in the reverse direction then generates an average of the two original samples, since the sample in domain B should be mapped back to both the first and second original samples in A. Pairwise distance constraints prevents this from happening.

**Self-distance Constraints**   Whether or not the distance constraint is more effective than the circularity constraint in recovering the alignment, the distance based constraint has the advantage of being one sided. However, it requires that pairs of samples are transfered at once, which, while having little implications on the training process as it is currently done, might effect the ability to perform on-line learning. Furthermore, the official CycleGAN [28] implementation employs minibatches of size one. We, therefore, suggest an additional constraint, which employs one sample at a time and compares the distances between two parts of the same sample.

Let $L, R : \mathbb{R}^{h \times w} \to \mathbb{R}^{h \times w/2}$ be the operators that given an input image return the left or right part of it. We define the following loss:

$$\mathcal{L}_{\substack{\text{self-} \\ \text{distance}}}(G_{AB}, \hat{p}_A) = \mathbb{E}_{x \sim \hat{p}_A} | \frac{1}{\sigma_A}(\|L(x) - R(x)\|_1 - \mu_A)$$
$$- \frac{1}{\sigma_B}(\|L(G_{AB}(x)) - R(G_{AB}(x))\|_1 - \mu_B)| \qquad (1)$$

where $\mu_A$ and $\sigma_A$ are the mean and standard deviation of the pairwise distances between the two halves of the image in the training set from domain $A$, and similarly for $\mu_B$ and $\sigma_B$, e.g., given the training set $\{x_j\}_{j=1}^n \subset B$, $\mu_B$ is precomputed as $\frac{1}{n} \sum_j \|L(x_j) - R(x_j)\|_1$.

## 3.1   Network Architecture and Training

When training the networks $G_{AB}, G_{BA}, D_B$ and $D_A$, we employ the following loss, which is minimized over $G_{AB}$ and $G_{BA}$ and maximized over $D_B$ and $D_A$:

$$\alpha_{1A} \mathcal{L}_{\text{GAN}}(G_{AB}, D_B, \hat{p}_A, \hat{p}_B) + \alpha_{1B} \mathcal{L}_{\text{GAN}}(G_{BA}, D_A, \hat{p}_B, \hat{p}_A) + \alpha_{2A} \mathcal{L}_{\text{cycle}}(G_{AB}, G_{BA}, \hat{p}_A) +$$
$$\alpha_{2B} \mathcal{L}_{\text{cycle}}(G_{BA}, G_{AB}, \hat{p}_B) + \alpha_{3A} \mathcal{L}_{\text{distance}}(G_{AB}, \hat{p}_A) + \alpha_{3B} \mathcal{L}_{\text{distance}}(G_{BA}, \hat{p}_B) +$$
$$\alpha_{4A} \mathcal{L}_{\text{self-distance}}(G_{AB}, \hat{p}_A) + \alpha_{4B} \mathcal{L}_{\text{self-distance}}(G_{BA}, \hat{p}_B)$$

where $\alpha_{iA}, \alpha_{iB}$ are trade-off parameters. We did not test the distance constraint and the self-distance constraint jointly, so in every experiment, either $\alpha_{3A} = \alpha_{3B} = 0$ or $\alpha_{4A} = \alpha_{4A} = 0$. When performing one sided mapping from $A$ to $B$, only $\alpha_{1A}$ and either $\alpha_{3A}$ or $\alpha_{4A}$ are non-zero.

We consider A and B to be a subset of $\mathbb{R}^{3 \times s \times s}$ of images where $s$ is either 64, 128 or 256, depending on the image resolution. In order to directly compare our results with previous work and to employ the strongest baseline in each dataset, we employ the generator and discriminator architectures of both DiscoGAN [11] and CycleGAN [28].

In DiscoGAN, the generator is build of an encoder-decoder unit. The encoder consists of convolutional layers with $4 \times 4$ filters followed by Leaky $ReLU$ activation units. The decoder consists of deconvolutional layers with $4 \times 4$ filters followed by a $ReLU$ activation units. Sigmoid is used for the output layer and batch normalization [8] is used before the $ReLU$ or Leaky $ReLU$ activations. Between 4 to 5 convolutional/deconvolutional layers are used, depending on the domains used in A and B (we match the published code architecture per dataset). The discriminator is similar to the encoder, but has an additional convolutional layer as the first layer and a sigmoid output unit.

The CycleGAN architecture for the generator is based on [10]. The generators consist of two 2-stride convolutional layers, between 6 to 9 residual blocks depending on the image resolution and two fractionally strided convolutions with stride $1/2$. Instance normalization is used as in [10]. The discriminator uses $70 \times 70$ PatchGANs [9]. For training, CycleGAN employs two additional techniques. The first is to replace the negative log-likelihood by a least square loss [25] and the second is to use a history of images for the discriminators, rather then only the last image generated [20].

Table 1: Tradeoff weights for each experiment.

| Experiment | $\alpha_{1A}$ | $\alpha_{1B}$ | $\alpha_{2A}$ | $\alpha_{2B}$ | $\alpha_{3A}$ | $\alpha_{3B}$ | $\alpha_{4A}$ | $\alpha_{4B}$ |
|---|---|---|---|---|---|---|---|---|
| DiscoGAN | 0.5 | 0.5 | 0.5 | 0.5 | 0 | 0 | 0 | 0 |
| Distance → | 0.5 | 0 | 0 | 0 | 0.5 | 0 | 0 | 0 |
| Distance ← | 0 | 0.5 | 0 | 0 | 0 | 0.5 | 0 | 0 |
| Dist+Cycle | 0.5 | 0.5 | 0.5 | 0.5 | 0.5 | 0.5 | 0 | 0 |
| Self Dist → | 0.5 | 0 | 0 | 0 | 0 | 0 | 0.5 | 0 |
| Self Dist ← | 0 | 0.5 | 0 | 0 | 0 | 0 | 0 | 0.5 |

Table 2: Normalized RMSE between the angles of source and translated images.

| Method | car2car | car2head |
|---|---|---|
| DiscoGAN | 0.306 | 0.137 |
| Distance | 0.135 | **0.097** |
| Dist.+Cycle | **0.098** | 0.273 |
| Self Dist. | 0.117 | 0.197 |

Table 3: MNIST classification on mapped SHVN images.

| Method | Accuracy |
|---|---|
| CycleGAN | 26.1% |
| Distance | **26.8%** |
| Dist.+Cycle | 18.0% |
| Self Dist. | 25.2% |

Table 4: CelebA mapping results using the VGG face descriptor.

| Method | Male → Female | | Blond → Black | | Glasses → Without | |
|---|---|---|---|---|---|---|
| | Cosine Similarity | Separation Accuracy | Cosine Similarity | Separation Accuracy | Cosine Similarity | Separation Accuracy |
| DiscoGAN | 0.23 | 0.87 | 0.15 | 0.89 | 0.13 | **0.84** |
| Distance | 0.32 | **0.88** | **0.24** | **0.92** | **0.42** | 0.79 |
| Distance+Cycle | **0.35** | 0.87 | **0.24** | 0.91 | 0.41 | 0.82 |
| Self Distance | 0.24 | 0.86 | **0.24** | 0.91 | 0.34 | 0.80 |
| ——————— Other direction ——————— | | | | | | |
| DiscoGAN | 0.22 | 0.86 | 0.14 | 0.91 | 0.10 | **0.90** |
| Distance | 0.26 | 0.87 | **0.22** | **0.96** | **0.30** | 0.89 |
| Distance+Cycle | **0.31** | 0.89 | **0.22** | 0.95 | **0.30** | 0.85 |
| Self Distance | 0.24 | **0.91** | 0.19 | 0.94 | **0.30** | 0.81 |

## 4 Experiments

We compare multiple methods: the DiscoGAN or the CycleGAN baselines; the one sided mapping using $\mathcal{L}_{distance}$ ($A \to B$ or $B \to A$); the combination of the baseline method with $\mathcal{L}_{distance}$; the self distance method. For DiscoGAN, we use a fixed weight configuration for all experiments, as shown in Tab. 1. For CycleGAN, there is more sensitivity to parameters and while the general pattern is preserved, we used different weight for the distance constraint depending on the experiment, digits or horses to zebra.

**Models based on DiscoGAN** Datasets that were tested by DiscoGAN are evaluated here using this architecture. In initial tests, CycleGAN is not competitive on these out of the box. The first set of experiments maps rotated images of cars to either cars or heads. The 3D car dataset [4] consists of rendered images of 3D cars whose degree varies at $15°$ intervals. Similarly, the head dataset, [17], consists of 3D images of rotated heads which vary from $-70°$ to $70°$. For the car2car experiment, the car dataset is split into two parts, one of which is used for A and one for B (It is further split into train and test set). Since the rotation angle presents the largest source of variability, and since the rotation operation is shared between the datasets, we expect it to be the major invariant that the network learns, i.e., a semantic mapping would preserve angles.

A regressor was trained to calculate the angle of a given car image based on the training data. Tab. 2 shows the Root Mean Square Error (RMSE) between the angle of source image and translated image. As can be seen, the pairwise distance based mapping results in lower error than the DiscoGAN one, combining both further improves results, and the self distance outperforms both DiscoGAN and pairwise distance. The original DiscoGAN implementation was used, but due to differences in evaluation (different regressors) these numbers are not compatible with the graph shown in DiscoGAN.

For car2head, DiscoGAN's solution produces mirror images and combination of DiscoGAN's circularity constraint with the distance constraint produces a solution that is rotated by $90°$. We consider these biases as ambiguities in the mapping and not as mistakes and, therefore, remove the mean error prior to computing the RMSE. In this experiment, distance outperforms all other methods. The combination of both methods is less competitive than both, perhaps since each method pulls toward a different solution. Self distance, is worse than circularity in this dataset.

Another set of experiments arises from considering face images with and without a certain property. CelebA [26, 14] was annotated for multiple attributes including the person's gender, hair color, and the existence of glasses in the image. Following [11] we perform mapping between two values of each of these three properties. The results are shown in the supplementary material with some examples in Fig. 3. It is evident that the DiscoGAN method (using the unmodified authors' implementation) presents many more failure cases than our pair based method. The self-distance method was implemented with the top and bottom image halves, instead of left to right distances, since faces are symmetric. This method also seems to outperform DiscoGAN.

In order to evaluate how well the face translation was performed, we use the representation layer of VGG faces [16] on the image in A and its output in B. One can assume that two images that match will have many similar features and so the VGG representation will be similar. The cosine similarities, as evaluated between input images and their mapped versions, are shown in Tab. 4. In all cases, the pair-distance produces more similar input-output faces. Self-distance performs slightly worse than pairs, but generally better than DiscoGAN. Applying circularity together with pair-distance, provides the best results but requires, unlike the distance, learning both sides simultaneously.

While we create images that better match in the face descriptor metric, our ability to create images that are faithful to the second distribution is not impaired. This is demonstrated by learning a linear classifier between the two domains based on the training samples and then applying it to a set of test image before and after mapping. The separation accuracy between the input test image and the mapped version is also shown in Tab. 4. As can be seen, the separation ability of our method is similar to that of DiscoGAN (it arises from the shared GAN terms).

We additionally perform a user study to asses the quality of our results. The user is first presented with a set of real images from the dataset. Then, 50 random pairs of images are presented to a user for a second, one trained using DiscoGAN and one using our method. The user is asked to decide which image looks more realistic. The test was performed on 22 users. On shoes to handbags translation, our translation performed better on 65% of the cases. For handbags to shoes, the score was 87%. For male to female, both methods showed a similar realness score (51% to 49% of DiscoGAN's). We, therefore, asked a second question: given the face of a male, which of the two generated female variants is a better fit to the original face. Our method wins 88% of the time.

In addition, in the supplementary material we compare the losses of the GAN discriminator for the various methods and show that these values are almost identical. We also measure the losses of the various methods during test, even if these were not directly optimized. For example, despite this constraints not being enforced, the distance based methods seem to present a low circularity loss, while DiscoGAN presents a relatively higher distance losses.

Sample results of mapping shoes to handbags and edges to shoes and vice versa using the DiscoGAN baseline architecture are shown in Fig. 3. More results are shown in the supplementary. Visually, the results of the distance-based approach seem better then DiscoGAN while the results of self-distance are somewhat worse. The combination of DiscoGAN and distance usually works best.

**Models based on CycleGAN** Using the CycleGAN architecture we map horses to zebras, see Fig. 4 and supplementary material for examples. Note that on the zebra to horse mapping, all methods fail albeit in different ways. Subjectively, it seems that the distance + cycle method shows the most promise in this translation.

In order to obtain numerical results, we use the baseline CycleGAN method as well as our methods in order to translate from Street View House Numbers (SVHN) [15] to MNIST [12]. Accuracy is then measured in the MNIST space by using a neural net trained for this task. Results are shown in Tab. 3 and visually in the Supplementary. While the pairwise distance based method improves upon the baseline method, there is still a large gap between the unsupervised and semi-supervised setting presented in [22], which achieves much higher results. This can be explained by the large amount of irrelevant information in the SVHN images (examples are shown in the supplementary). Combining the distance based constraint with the circularity one does not work well on this dataset.

We additionally performed a qualitative evaluation using FCN score as in [28]. The FCN metric evaluates the interoperability images by taking a generated cityscape image and generating a label using semantic segmentation algorithm. The generated label can then be compared to the ground truth label. FCN results are given as three measures: per-pixel accuracy, per-class accuracy and Class

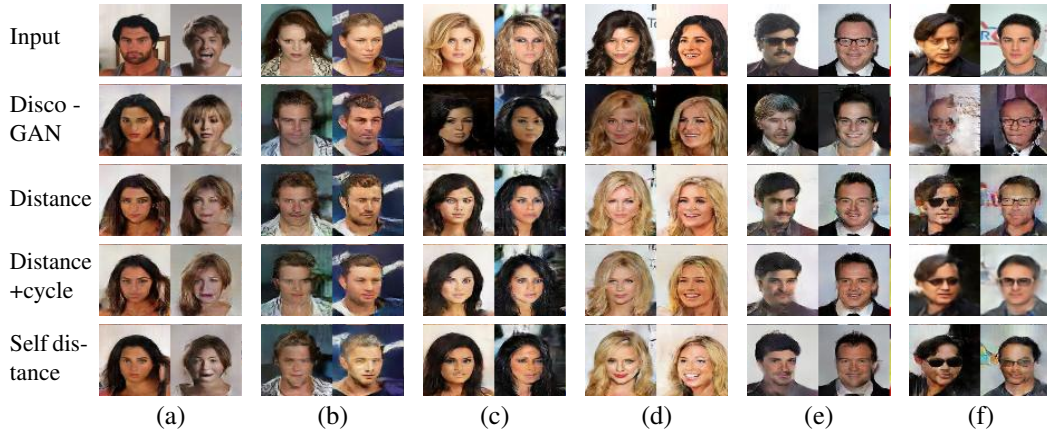

Figure 3: Translations using various methods on the celebA dataset: (a,b) Male to and from Female. (c,d) Blond to and from black hair. (e,f) With eyeglasses to from without eyeglasses.

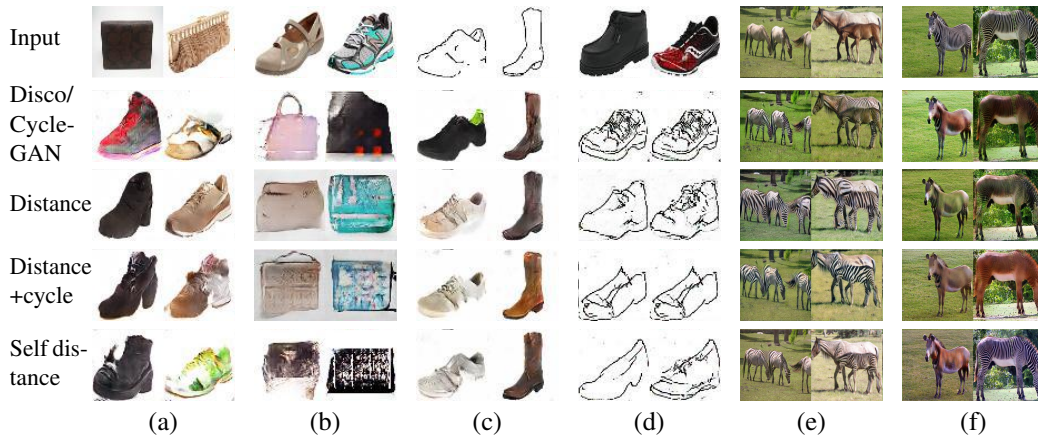

Figure 4: (a,b) Handbags to and from shoes. (c,d) Edges to/from shoes. (e,f) Horse to/from zebra.

IOU. Our distance GAN method is preferable on all three scores (0.53 vs. 0.52, 0.19 vs. 0.17, and 0.11 vs 0.11, respectively). The paired $t$-test $p$-values are $0.29$, $0.002$ and $0.42$ respectively. In a user study similar to the one for DiscoGAN above, our cityscapes translation scores 71% for realness when comparing to CycleGAN's. When looking at similarity to the ground truth image we score 68%.

# 5    Conclusion

We have proposed an unsupervised distance-based loss for learning a single mapping (without its inverse), which empirically outperforms the circularity loss. It is interesting to note that the new loss is applied to raw RGB image values. This is in contrast to all of the work we are aware of that computes image similarity. Clearly, image descriptors or low-layer network activations can be used. However, by considering only RGB values, we not only show the general utility of our method, but also further demonstrate that a minimal amount of information is needed in order to form analogies between two related domains.

# Acknowledgements

This project has received funding from the European Research Council (ERC) under the European Union's Horizon 2020 research and innovation programme (grant ERC CoG 725974). The authors would like to thank Laurens van der Maaten and Ross Girshick for insightful discussions.

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
