[Supplementary Material]

# Supplementary Material for:
# One-Sided Unsupervised Domain Mapping

## 1 Experiments with the DiscoGAN architecture

Tab. 1 presents the eight losses measured for each of the methods. For example, we can measure the distance loss for the DiscoGAN method even tough it is not part of its loss. To allow the computation of circularity, here, the distance method was run in both directions at once.

Table 1: Losses measured for each method on the CelebA dataset.

| Method | $\mathcal{L}_{\text{GAN}}(A)$ | $\mathcal{L}_{\text{GAN}}(B)$ | $\mathcal{L}_{\text{cycle}}(B)$ | $\mathcal{L}_{\text{cycle}}(B)$ | $\mathcal{L}_{\text{dist}}(A)$ | $\mathcal{L}_{\text{dist}}(B)$ | $\mathcal{L}_{\text{selfd}}(A)$ | $\mathcal{L}_{\text{selfd}}(B)$ |
|---|---|---|---|---|---|---|---|---|
| (A) Male to (B) Female: | | | | | | | | |
| DiscoGAN | 4.300 | 2.996 | 0.036 | 0.024 | 0.466 | 0.457 | 0.441 | 0.422 |
| Distance | 3.702 | 2.132 | 0.026 | 0.026 | 0.047 | 0.047 | 0.038 | 0.044 |
| Distance+Cycle | 4.280 | 1.651 | 0.017 | 0.016 | 0.046 | 0.043 | 0.042 | 0.040 |
| Self Distance | 3.322 | 3.131 | 0.092 | 0.091 | 0.048 | 0.050 | 0.045 | 0.044 |
| (A) Blond to (B) Black hair: | | | | | | | | |
| DiscoGAN | 2.511 | 3.297 | 0.019 | 0.018 | 0.396 | 0.399 | 0.396 | 0.399 |
| Distance | 0.932 | 2.243 | 0.021 | 0.017 | 0.046 | 0.042 | 0.046 | 0.042 |
| Distance+Cycle | 1.045 | 2.484 | 0.013 | 0.012 | 0.043 | 0.043 | 0.043 | 0.042 |
| Self Distance | 0.965 | 2.867 | 0.022 | 0.018 | 0.049 | 0.048 | 0.049 | 0.048 |
| (A) With or (B) Without eyeglasses: | | | | | | | | |
| DiscoGAN | 5.734 | 3.621 | 0.110 | 0.040 | 0.535 | 0.337 | 0.535 | 0.074 |
| Distance | 7.697 | 0.804 | 0.046 | 0.036 | 0.023 | 0.065 | 0.023 | 0.065 |
| Distance+Cycle | 5.730 | 0.924 | 0.024 | 0.017 | 0.027 | 0.048 | 0.028 | 0.048 |
| Self Distance | 8.242 | 0.795 | 0.040 | 0.018 | 0.029 | 0.051 | 0.029 | 0.050 |

Fig. 1, 2, 3, 4, 5 present images from multiple mapping experiments that employ the same network architecture as DiscoGAN.

(a) Input

(b) Disco-GAN

(c) Distance

(d) Cycle+dist

(e) Self-distance

(Male to female)　　　　　　　(Female to male)

Figure 1:　Results for celebA Male to Female transfer (a) Input (b) DiscoGAN model. (c) Distance model (our model) trained with A and B simultaneously. (d) DiscoGAN and Distance model. (e) Distance model where distances are compared within the image s.t the distance from top half to bottom half is preserved.

(a) Input

(b) Disco-GAN

(c) Distance

(d) Cycle+dist

(e) Self-distance

(Blond to black hair)　　　　　　　(Black to blond hair)

Figure 2:　Same as Fig. 1 but with black to blond hair conversion

(a) Input

(b) Disco-GAN

(c) Distance

(d) Cycle+dist

(e) Self-distance

(With to without eyeglasses)　　　　(Without to with eyeglasses)

Figure 3:　Same as Fig. 1 but with eyeglasses to no eyeglasses and no eyeglasses to eyeglasses conversion.

(a) Input

(b) Disco-GAN

(c) Distance

(d) Cycle+dist

(e) Self-distance

(Handbags to shoes)　　　　(Shoes to handbags)

Figure 4:　Same as Fig. 1 but with handbags to shoes and shoes to handbags conversion.

(a) Input

(b) Disco-GAN

(c) Distance

(d) Cycle+dist

(e) Self-distance

(Edges to shoes)          (Shoes to edges)

Figure 5: Same as Fig. 1 but edges to shoes and shoes to edges conversion.

# 2 Experiments with the CycleGAN architecture

Fig. 6 gives a translation between images of horses to zebra. Fig. 7 gives a translation between images of zebra to horse. Fig. 8 presents examples of transforming SVHN images to MNIST digits. Fig. 9 gives a translation from cityscapes labels to images.

Figure 6: Translation from horse to zebra based on the CycleGAN architecture.

(a) Input

(c)Cycle-GAN

(d) Distance

(e) Cycle+dist

(f) Self-distance

Figure 7: Translation from zebra to horse based on the CycleGAN architecture.

Figure 8: Translating SVHN to MNIST with a CycleGAN architecture

(a) Input

(c)Cycle-GAN

(d) Distance

Figure 9: Translation from labels to cityscapes images based on the CycleGAN architecture.