[Reviews · NeurIPS 2017]

Reviewer 1



This paper tackles the problem of unsupervised domain adaptation. The paper introduces a new constraint, which compares samples and enforces high cross-domain correlation between the matching distances computed in each domain. An alternative to pairwise distance is provided, for cases in which we only have access to one data sample at a time. In this case, the same rationale can be applied by splitting the images and comparing the distances between their left/right or up/down halves in both domains. The final unsupervised domain adaptation model is trained by combining previously introduced losses (adversarial loss and circularity loss) with the new distance loss, showing that the new constraint is effective and allows for one directional mapping. The paper is well structured and easy to follow. The main contribution is the introduction of a distance constraint to recent state-of-the-art unsupervised domain adaptation pipeline. The introduction of the distance constraint and the assumption held by this constraint, i.e. there is a high degree of correlation between the pairwise distances in the 2 domains, are well motivated with experimental evidence. Although literature review is quite extensive, [a-c] might be relevant to discuss. [a] https://arxiv.org/pdf/1608.06019.pdf [b] https://arxiv.org/pdf/1612.05424.pdf [c] https://arxiv.org/pdf/1612.02649.pdf The influence of having a distance loss is extensively evaluated on a variety of datasets for models based on DiscoGAN, by changing the loss function (either maintaining all its components, or switching off some of them). Qualitative and quantitative results show the potential of the distance loss, both in combination with the cycle loss and on its own, highlighting the possibility of learning only one-sided systems. However, given that the quantitative evaluation pipeline is not robust (i.e. it depends on training a regressor), it is difficult to make any strong claims on the performance of the method. The authors qualitatively and quantitatively assess their contributions for models based on CycleGAN as well. Quantitative results are showed for SVHN to MNIST mapping. However, among the large variety of mappings shown in the CycleGAN paper, authors only pick the horse to zebra mapping to make the qualitative comparison. Given the subjectivity of this kind of comparison, it would be more compelling if the authors could show some other mappings such as season transfer, style transfer, other object transfigurations or labels2photo/photo2label task. Quantitative evaluation of CycleGAN-based models could be further improved by following the FCN-scores of the CycleGAN paper on the task labels/photo or photo/labels. Finally, in the conclusions, the claim “which empirically outperforms the circularity loss” (lines 293-294) seems a bit too strong given the experimental evidence.

Reviewer 2



This paper introduces one-sided unsupervised domain mapping that exploits smoothing assumption (images that are similar in domain A should be similar in domain B). The paper is well written and easy to follow. The contribution of the paper is based on simple observation that allows the authors to perform domain mapping without performing cycles (A->B->A). The empirical results seem to be pointing that the smoothness assumption itself leads to good results. Table 2: In DiscoGan paper the reported RMSE for car2car task is 0.11 (see Fig. 5(c) in https://arxiv.org/pdf/1703.05192.pdf), the authors for the same task report the result of 0.306. What might be the reason of such big discrepancies in the reported scores? Is it only due to regressor (as mentioned in line242)? Comparison with CycleGAN. Would it be possible to show more qualitative results on different tasks used in CycleGAN paper (e. g. label to photo)? Line 228: “Datasets that were tested by DiscoGAN are evaluated here using this architecture” -> please rephrase. Line 298: “We have proposed an unsupervised distance-based loss for learning a single mapping (without its inverse) and which empirically outperforms the circularity loss.” I’m not sure the experimental evidence presented in the paper is sufficient to support this claim.

Reviewer 3



This paper introduces a novel loss to train an image-to-image mapping from unpaired data. The idea is that there is a high correlation between the pairwise distances in the source and target domains. These distances can be measured between two different samples or two regions in the same sample. Positive: - Using pairwise distances in this context is new and interesting. Negative: - It seems to me that, while the notion of pairwise distances can indeed be well-suited in some cases, it is much less so in other ones. For example, two images depicting the same object but in different colors will be very different, but the corresponding edge images would look very similar. Similarly, two Street View House images depicting the same number but in different colors will look different, but should correspond to very similar MNIST images. In essence, while the statement on Line 86 is correct, the reverse is not necessarily true. - In Section 2 (Fig. 1), the authors perform different motivating experiments using the results on the CycleGAN. In my opinion, these experiments would better motivate the method if they were done on data with known pairs. These results might just come from artefacts of the CycleGAN. - I am surprised that the one-sided version of this method works. In essence, the distance-based loss on Page 5 does not really make use of the target data, except in the mean \mu_B and std \sigma_B. While this may already contain important information for a domain such as object edges, it seems that it would throw away much information for other domains. Can the authors comment on this? - Although [18] is discussed as a weakly-supervised method, I believe that it can apply quite generally. For examples, the features of deep networks, such as the VGG network, trained on ImageNet have proven effective in many tasks. By using such features, the weak supervision would come at virtually no cost. I would suggest the authors to compare their results with [18] based on such features. The corresponding loss could also be added to the different ones used in Section 4. - It is not always clear that the metrics used for quantitative evaluation (based on additional regressor, classifier, VGG features, or directly on loss values as in the supplementary material) really reflect human perception. For example, in the supplementary material, I find the results of DiscoGAN in Fig. 1 more convincing than those of the proposed method. In my opinion, a user study would be more valuable. - The introduction makes it sound like this is the first method to work in the unsupervised scenario (Lines 28-29). It becomes clear later that the authors do not mean to claim this, but I think it should be clarified from the beginning.